# Multicomponent Intervention Associated with Improved Emotional and Cognitive Outcomes of Marginalized Unemployed Youth of Latin America

Cristina Crespo-Andrade [1,*], Ana Francisca Trueba [1,2,3], María Sol Garcés [1,3] and Graham Pluck [3,4]

1   Colegio de Ciencias Sociales y Humanidades, Universidad San Francisco de Quito, Quito 170901, Ecuador; atrueba@usfq.edu.ec (A.F.T.); sgarces@usfq.edu.ec (M.S.G.)
2   Department of Psychiatry, Harvard Medical School, Boston, MA 02115, USA
3   Instituto de Neurociencias, Universidad San Francisco de Quito, Quito 170901, Ecuador; graham.ch@chula.ac.th
4   Faculty of Psychology, Chulalongkorn University, Bangkok 10330, Thailand
*   Correspondence: mcrespoa@usfq.edu.ec

**Abstract:** Mass migration and people seeking political refuge are critical social issues facing Latin America. Ecuador has the largest population of recognized refugees in the region. Youths from a migration background have an increased risk of becoming NEET (Young people not in employment, education, or training). Such youths struggle more with mental health problems than non-NEET peers. Being a refugee, NEET further increases the risk of having mental health problems and may be linked to lower cognitive functioning, which could maintain exclusion and unemployment. This intervention study was performed with a group of young people of different nationalities who were refugees or belonged to other vulnerable groups attending a six-week employability-support intervention in Ecuador. In order to assess the impact of the intervention, a range of measures of executive cognitive function, mental health, and the potential for positive change were used. We found that post-intervention, the group reported significantly less psychological distress and better self-esteem, self-efficacy, and cognitive response inhibition than before the intervention. We conclude that multicomponent interventions may effectively improve the psychological functioning of vulnerable NEET groups in the Latin American context.

**Keywords:** NEETS; refugees; mental health; emotion regulation; self-efficacy; cognitive abilities; executive function; employability support; intervention; Latin America

## 1. Introduction

Education and employment are positively related to cognitive and mental health (Chen et al. 2019; Vance et al. 2016; Hergenrather et al. 2018; Modini et al. 2016). Participating in education and employment requires learning new skills, establishing goals and routines, and engaging socially, benefiting well-being by creating occupational and social support and enhancing cognition (Vance et al. 2016; O'Dea et al. 2014). Dropout from education or disengagement of the labor market exposes youth to a greater risk of social exclusion, psychological distress, disability, maladaptive behaviors, illness, and disease (O'Dea et al. 2014; Fernández-Suárez et al. 2016; Hjorth et al. 2016).

According to the International Labor Organization, there is a high rate of young people, more than one in five (22.4%) aged 15–24, neither in employment, education, nor training (NEET). Acknowledging this problem worldwide led to adopting the NEET rate as part of the 2030 Agenda for Sustainable Development, as an indicator of progress towards reducing the proportion of the NEET individuals within a population (O'Higgins 2020; Organisation for Economic Co-Operation and Development et al. 2016; Organisation for Economic Co-Operation and Development 2021).

NEET rates vary widely across countries, showing the highest rates in lower-middle-income countries and with young women outnumbering young men by a factor of two to one (O'Higgins 2020). NEETs are not a homogeneous group; the causes behind this status are diverse (Tamesberger and Bacher 2014). The Eurofound (2012, 2016) suggested subgroups in the NEET population that include a mix of vulnerable and non-vulnerable young people with different characteristics and needs. The individual risk factors include adverse family environments, low household income or educational levels, disability, living in remote areas, and immigration background (Eurofound 2012; Assmann and Broschinski 2021).

### 1.1. NEET Mental Health and Cognition

Evidence suggests that NEETs struggle more with mental health problems than non-NEET peers. NEETs may be at greater risk of depression and anxiety (Feng et al. 2015) and severe symptoms of these disorders (Basta et al. 2019; O'Dea et al. 2014). The relationship between mental health and NEET status could be bidirectional: the pre-existence of a mental health condition may constitute a risk factor for becoming NEET, and conversely, mental health problems could be a long-term consequence of NEET status. In a British cohort study, nearly 60% of NEETs had suffered from a mental health problem during childhood or adolescence compared to 35% of non-NEETs (Goldman-Mellor et al. 2015). Moreover, an association between NEET status during adolescence and several mental health outcomes in adulthood is established (Gutiérrez-García et al. 2017; Power et al. 2015). The mental health problems experienced by NEETS are thought to have been exacerbated by the effects of the COVID-19 pandemic (Kvieskienė et al. 2021).

Cognitive and non-cognitive skills negatively correlate with the probability of being NEET. Higher scores in cognitive measures and socio-emotional abilities such as self-esteem, self-efficacy, and locus of control are associated with a lower probability of being NEET (Alvarado et al. 2020). Education has been related to better performance on cognitive measures and further engagement with cognitive activities and challenges (Lövdén et al. 2020; Guerra-Carrillo et al. 2017). Parisi et al. (2012) found that educational and other cognitively charged experiences may benefit brain structure and function, developing neuroprotective effects to face age-related decline. Additionally, education has economic implications as cognitive and non-cognitive skills have labor-market value and impacts individual earnings (De Hoyos et al. 2016). Therefore, the lack of opportunities to develop skills is crucial for future NEET status (Gladwell et al. 2015).

### 1.2. NEETs in Ecuador

Like other countries in Latin America, Ecuador has a high percentage of people who could be defined as NEET. The total number of people between 15 and 24 years of age who are NEET in Latin America remained practically constant between the beginning and the end of 1992–2014: it went from 19.0 million in 1992 to 18.7 million in 2014 (Tornarolli 2016). Four out of five NEETs in Latin America are females; they more often come from a rural background and belong to the poorest quintile of the wealth distribution (Minujín et al. 2016). Among young Ecuadorians aged between 15 and 24, 19% do not attend any educational center and are unemployed (Buitrón et al. 2018).

### 1.3. Refugees in Ecuador

As mentioned earlier, being a refugee is a risk factor for becoming NEET. According to the United Nations High Commissioner for Refugees (2020), Ecuador has the largest population of recognized refugees in Latin America, with 69,897 recognized refugees arriving between 1989 and 2020, most of them coming from Colombia and Venezuela. For decades, Colombians were forced to leave their country, escaping the violence of the internal conflict and looking for security in Ecuador (United Nations High Commissioner for Refugees 2020). In 2019, 79.5 million people were forcibly displaced worldwide. Four million of them left Venezuela and now live mainly in Latin America and the Caribbean (United Nations High

Commissioner for Refugees 2020). In the first quarter of 2019, around 470,095 Venezuelans lived in Ecuador, almost 3% of Ecuador's population (Olivieri et al. 2020).

For both groups, but especially for Venezuelans, forced migration increased the vulnerability to 'labor and sexual exploitation, trafficking, violence, discrimination and xenophobia' (United Nations High Commissioner for Refugees 2021). Refugees and asylum seekers are among the most vulnerable groups (United Nations High Commissioner for Refugees 2015, 2016). Their vulnerability is linked to experiences before migration due to conflict, violence, or persecution and to post-migration stressors such as exclusion, unemployment, and discrimination (Hameed et al. 2018). As reported in the literature, having a migration background increases the youth person's likelihood of becoming NEET by 70% (Eurofound 2012). Therefore, the probability that a significant percentage of young refugees in Ecuador are in NEET status is high, becoming a socioeconomic concern.

### 1.4. Mental Health and Cognition of Refugees

The adverse psychosocial factors experienced during or after migration could negatively impact migrant mental-health status and increase the risk of cognitive impairment (Xu et al. 2018). Most of the current research on the mental health of refugees has shown that they are at higher risk for a variety of psychiatric disorders such as post-traumatic stress disorder, anxiety, and depression, due to their experiences before migration (Peterson et al. 2020; Chaplin et al. 2020; Hameed et al. 2018). In addition, post-migration distress, mainly acculturative stress, can worsen mental health symptoms (Hameed et al. 2018).

There is strong evidence that cognitive factors are essential for successful refugee adjustment (Hahn et al. 2019). Xu et al. (2018) identify different pathways that may link migration and cognitive function from current evidence. Socioeconomic status, psychosocial and behavioral traits, and physical and psychological health prior to migration could influence cognitive function (Xu et al. 2018). A study comparing executive cognitive abilities, specifically working memory and inhibitory control, between refugee and non-refugee adolescents, found that poverty was more critical to cognitive function than traumatic experiences (Chen et al. 2019). Therefore, since dropout from education and unemployment are risk factors for poverty, refugees in the NEET situation may also be more susceptible to executive cognitive problems. Executive functions are important considerations because they are both significant predictors of educational and workplace success (Pluck et al. 2019; Pluck et al. 2020b) and are also known to be malleable, being sensitive to behavioral interventions (Stamenova and Levine 2018). For instance, in previous studies, we found that the Hayling Test that measures verbal response suppression was the best predictor of workplace success, specifically sales (Pluck et al. 2020b) as well a predictor of academic performance in college and high school (Pluck et al. 2019). Cognitive switching measured using Trail Making Test of the Delis–Kaplan Executive Function System was a predictor of working hours in populations struggling with mental (McGurk and Mueser 2006) and physical health (Rabkin et al. 2004), and spatial working memory is a predictor of vocational participation (Cairns et al. 2017).

### 1.5. Relevance

The population examined in this study is doubly vulnerable since they are migrants and have NEET status. As mentioned previously, being a refugee NEET increases the risk of having mental health problems and may impact cognitive ability. Interventions that improve cognitive functioning and mental health may be important ways to break the cycle of social exclusion. Furthermore, in addition to having worse mental health, a study conducted by Buckman et al. (2021) suggests that NEETs had less successful outcomes following psychological interventions in terms of recovery, deterioration, and attrition, indicating that they need early intervention and more sessions.

*1.6. Aims of the Study*

Previous research has shown that young refugees need diverse activities and practices to promote their socioeconomic inclusion (Pastoor 2017). This study aimed to assess the impact of one such multicomponent employability-support intervention. We assessed the effect of the intervention on the mental health and cognitive abilities of a group of refugees who were in the NEET status in Ecuador (aged 18–36). Our first hypothesis was that our intervention would result in significant improvements in a range of emotional factors such as less psychological distress (General Health Questionnaire-28; GHQ-28), fewer difficulties in emotion regulation (Difficulties in Emotion Regulation Scale total score; DERS), higher self-efficacy (Generalized Self-efficacy Scale; GSE), higher self-esteem (Rosenberg Self-esteem Scale; RSES), higher locus of control (Rotter's Locus of Control Scale; LoC), and lower depression and anxiety (Hospital Anxiety Depression Scale; HADS). Our second hypothesis was that our intervention would result in significant improvements in executive function, specifically working memory, cognitive switching, and response suppression. We selected the Hayling Sentence Completion Test to evaluate response suppression, the Trails Making Test to measure cognitive switching, and finally, the Counting Span Test to evaluate working memory, as previous research suggests that these cognitive measures are predictors of academic (Pluck et al. 2019) and workplace success (Cairns et al. 2017; McGurk and Mueser 2006; Pluck et al. 2020b; Rabkin et al. 2004).

## 2. Material and Methods

*2.1. Research Design*

This was an opportunistic study on a group of adults (aged 18–36) undertaking a six-week-long psychological intervention to assist them in finding employment or further study. A pre-and post-intervention assessment was employed. The study focused on psychological distress and measures related to self-esteem, self-efficacy, and executive cognitive functioning.

*2.2. Participants*

Fifty-five people took part in the intervention. Of those 55, 51 (93%) consented to be part of the research study. As this was an opportunistic study, no predetermination of sample size was made; nevertheless, it was anticipated that the recruited sample would be sufficient to detect medium-sized effects at 80% power (Cohen 1992).

*2.3. The Intervention*

This intervention took place in 2018, and it lasted six weeks. 'A Ganar: Sin Limites' is an outreach project with the collaboration of three partners: FUDELA (Fundación de las Americas), United Nations High Commissioner for Refugees, and Universidad San Francisco de Quito.

FUDELA is a private non-profit organization that incorporates sport as a comprehensive training tool. The foundation works to guarantee children, adolescents, young people, and adults in the fulfillment of their rights in recreational, educational, and labor matters, preventing risk situations. FUDELA works in partnership with the United Nations High Commissioner for Refugees, who acts as the angel funder of the foundation's various programs.

The aim of "A Ganar: Sin Límites" is to provide skills to facilitate entrepreneurship opportunities and employability tools, as well as opportunities to access higher education. The target population is young people of different nationalities who were refugees or belonged to other vulnerable groups. During the summer of 2018, 55 participants from Colombia, Venezuela, Ecuador, and El Salvador were part of the project. In addition, one of the funder conditions was that the project included young Ecuadorian NEETs.

FUDELA attracts participants through social media and information provided by other organizations that work with refugees. The participants completed a survey and then participated in an interview. The project was over a six-week period, comprising 30 days of

activities, and took place at the Universidad San Francisco de Quito campus in Cumbayá, Ecuador. The participants were divided into two groups and attended different classes and activities. Smaller groups (around 15 participants) were formed for the sports activities and the psychology classes. The intervention, described in Table 1, had 11 modules, with a total of 235 h of activities, with full-time dedication. The intervention lessons and activities were run by faculty members of various departments of the university and assisted by students.

**Table 1.** Module contents and descriptions of the intervention.

| Module | Topics | Number of Hours |
| --- | --- | --- |
| Industrial quality and productivity | Quality, safety, use of equipment, basic tools, and the 5 S's principle. | 15 |
| Industrial techniques | Knowledge in the field of industrial technology to be able to solve basic problems. | 15 |
| Sales and customer service | Techniques to optimize sales, sales process, verbal, and non-verbal communication techniques. | 15 |
| Sports for development | Gamification using sports to develop communication, discipline, respect, teamwork and focus on results | 80 |
| Audiovisual production | Production of animated shorts and mini documentaries about the program. | 16 |
| Financial education | Financial education for small business: accounting, financial statements, tax, and cash flow, NPV, IRR and other investment rules in all their applications. | 15 |
| Services for hotels and restaurants | New cooking techniques and methods, preparation of a "mise en place", manipulation of raw materials. | 15 |
| Entrepreneurship | Business Model Canvas as a tool for the beneficiaries to evaluate business proposals and develop the added value of their venture. | 15 |
| Communication | Personal communication and organizational communication (managing social networks for a business, skills in their verbal and non-verbal communication). | 15 |
| Psychology | Activities focused on self-development, self-esteem, personal skills, introspection, resilience, limiting beliefs, how to stablish healthy limits and take care of oneself. | 25 |
| Job search skills | Preparation for job interviews, including non-verbal language, clothing, tone, and intonation of the voice. Guidance on design and updating of curriculum vitae. | 9 |

5 S's: Sort, Set in order, Shine, Standardize, Sustain. NPV: Net present value, IRR: Internal rate of return.

*2.4. Assessments*

2.4.1. Self-Report Measures

Six different questionaries were included in the current study. These were selected to assess a range of factors related to mental health and the potential for positive change.

A validated Spanish-language version (Lobo et al. 1986) of the General Health Questionnaire-28 (GHQ-28; Goldberg and Hillier 1979) was included and was graded using the Goodchild and Duncan-Jones (1985) method. Although subscale scores are

available, total scores across all items are typically used to quantify psychological distress. The GHQ-28 has been shown to have 'very good' internal consistency in an Ecuadorian non-clinical adult sample, with α values ranging from 0.81 to 0.87 (Pluck et al. 2020c). An additional measure of psychological distress was included, the Hospital Anxiety and Depression Scale (HADS; Zigmond and Snaith 1983), in a validated Spanish-language version (Herrero et al. 2003). Again, this is scored on a Likert scale, from 0 to 3 points. We found a 'reasonable' level of internal consistency for the HADS total score, at α = 0.73, from a previously reported Ecuadorian non-clinical adult sample (Pluck et al. 2020a). As with the GHQ-28, the HADS has subscale scores available, but research suggests it is essentially unidimensional, measuring psychological distress (Pallant and Tennant 2007). We, therefore, analyzed only the total scores from the GHQ-28 and HADS to test our principal hypotheses regarding mental health. Higher scores on both the GHQ-28 and the HADS indicate greater psychological distress.

We included a validated Spanish-language version (Hervás and Jódar 2008) of the 36-item Difficulties in Emotion Regulation Scale (DERS; Gratz and Roemer 2004). This is measured on a five-point Likert scale from 1–5 points for a range of different problems, including lack of awareness of emotional states, problems applying appropriate strategies to alleviate negative emotions, and difficulties controlling one's behavior during strong emotions. The total score appears appropriate for use in the Ecuadorian context, where it has a 'very good' internal consistency with α at 0.90 (Reivan-Ortiz et al. 2020). We, therefore, used the total DERS score to test our principal hypothesis regarding emotional regulation skills. Higher scores indicate greater difficulty with emotion regulation.

To measure self-esteem, we used a validated Spanish-language version (Torres 1983) of the Rosenburg Self-esteem Scale (RSES; Rosenberg 1965). This ten-item scale is scored on a Likert scale, from 1 to 4 points. As with other versions of the RSES, in an Ecuadorian adult sample, it was found to be essentially unidimensional, with 'very good' internal consistency at α = 0.84 (Bueno-Pacheco et al. 2020). We, therefore, used the total score to test our principal hypothesis regarding self-esteem. Higher scores on the RSES indicate higher self-esteem. We also used a validated Spanish-language version (Baessler and Schwarzer 1996) of the Generalized Self-efficacy Scale (GSE; Schwarzer and Jerusalem 1995). This is a 10-item questionnaire scored on a Likert scale, from 0 to 3 points. It has 'very good' internal consistency in the Ecuadorian context, with α at 0.91 (Bueno-Pacheco et al. 2018).

The final questionnaire used in this study was a Spanish version (Perez García 1984) of Rotter's Locus of Control Scale (LoC; Rotter 1966).

### 2.4.2. Executive Function Tests

Three different measures of higher-order cognitive control were employed in this study, based on the findings of Miyake et al. (2000) that there are three basic, somewhat independent, components of executive function. These are working memory, cognitive switching, and response suppression. In addition, all executive tasks employed here were language-based, as such functions are those most susceptible to socioeconomic deprivation (Pluck et al. 2021) and thus may have the greatest potential for remediation.

To measure working memory, we used the Counting Span Test (Conway et al. 2005). Ten separate trials were completed, graded from spans of two items to spans of five items.

To measure cognitive switching, we used the Trail Making Test of the Delis–Kaplan Executive Function System (Delis et al. 2001). This involves two baseline trials, one connecting numbers on an A3-size page and the other connecting letters. A switching trial is also completed in which the participant must alternate between connecting numbers and letters. The crucial statistic is the ratio of completion time on the switching trial to the average completion time on the baseline conditions, as this provides a relatively pure index of cognitive switching (Arbuthnott and Frank 2000). That ratio was used to test our principal hypothesis regarding this aspect of executive function.

To measure response suppression, we used the Hayling Sentence Completion Test (Burgess and Shallice 1996). We used only Part B, which is the test of response suppression,

and for convenience, we will refer to this simply as 'the Hayling'. Fifteen sentences are read aloud to the participant. However, for each sentence, the final word is missing, and the task of the participant is to quickly complete the sentence with a word that makes no sense (i.e., they must suppress the natural response to complete the utterance sensibly). Errors are recorded and scored from 0 points (good performance, a completely unrelated word) to 3 points (bad performance, a word that completes the utterance sensibly). Therefore, the possible score range is 0 to 45, with higher scores indicating greater difficulty with verbal response suppression. Response times for each sentence completion are also recorded.

We used two different Spanish-language versions of the Hayling, one created for research in response to suppression conducted in Cuba (Obeso et al. 2011) and the other created for use in Spain (Pérez-Pérez et al. 2016). In that latter study, the internal consistency of the error score was calculated and found to be 'very good' with α at 0.84. Therefore, that error score was used as the principal measure to assess our hypothesis regarding verbal response suppression as an executive function. The two different versions of the Hayling are equivalent, and both were used in this study so that different stimuli materials could be applied at pre-and post-intervention assessments (counterbalanced).

### 2.4.3. Institutional Review Board Statement

The Bioethics Committee of Universidad San Francisco de Quito reviewed the protocol, including the pre- and post-intervention assessments (application code: 2018-098IN, approval granted 9 May 2018). Written informed consent was taken from all participants who wished to take part. The research was conducted following all local laws and guidance on research provided by the American Psychological Association and by the Declaration of Helsinki of 1975, revised in 2013.

### 2.4.4. Procedure

All the participants of the intervention attended the campus of Universidad San Francisco de Quito, Ecuador. On the first day, the research aspect was introduced to them in a group meeting, and they were able to ask questions. Then consent forms were circulated. Those who consented were individually assessed on the three different executive function tests in one-to-one sessions with research assistants. The questionaries were distributed in booklet format and completed in a group setting. This was all completed on the first morning of the intervention.

The intervention then began. It lasted for six weeks and comprised of 30 individual days of intervention activities. On the final day, all participants who consented were assessed a second time, using the same questionnaires and cognitive tests. Participants were thanked for their participation in the research and debriefed about the nature of the research. They were able to ask questions and feedback to the researchers at this point. No incentive was given for participation in the research aspect of the intervention.

### 2.4.5. Statistical Methods

Raw data were used, processed, and analyzed with SPSS version 23. Data were summarized as mean and standard deviation. Parametric inferential analysis was by repeated-measures ANOVA, with effect sizes given as Cohen's d, calculated as the difference between pre- and post-intervention means divided by the pooled standard deviation (Cohen 1992). Qualitative interpretations of effect sizes are also based on Cohen (1992). For all ANOVA calculations, the residuals were considered normally distributed if the z scores for both skew and kurtosis were below 1.96 (Kim 2013). When non-normal distributions of residuals were observed, raw data were transformed and reanalyzed. When transformation could do produce normally distributed residuals, non-parametric tests were substituted. All statistical null-hypothesis tests were two-tailed. A significance threshold of 0.05 was selected.

## 3. Results

### 3.1. Demographic Information of the Sample

Of the 51 participants who completed the study, data were lost on basic demographics from a small proportion. Of those with data available, 30/49 (61%) identified as women, and the mean age was 24.47 (SD = 4.17, range = 18–36, *n* = 49). Notably, this is a somewhat higher age range than usually defined as NEET (i.e., 18–24), with 19/49 (39%) of our sample being older than 24. For this reason, additional analyses are performed on the main outcome variables when limiting the sample to those aged < 25. Of the 44 cases with ethnicity information available, the majority, 38/44 (86%), described themselves as mestizo (meaning mixed Amerindian and European ancestry). Of the others, 2/49 (5%) self-described as Black, 2/49 (5%) as mixed-race (4%), and 2/49 (5%) as White. The majority, 35/49 (71%), were migrants to Ecuador; the most common origin countries were Venezuela (*n* = 21) and Colombia (*n* = 12). All were Spanish speakers and were unemployed and not in education at the time of the intervention.

As we can identify two broad groups within the sample, Ecuadorian nationals (*n* = 14, 28%) and immigrants to Ecuador (*n* = 35, 72%), these were compared for both demographic and psychological status variables (i.e., scores on the questionnaires and cognitive tests, pre-intervention). These data are shown in Table 2, where it can be seen that the two sub-groups were similar on demographic variables, with no significant differences. Nor were there any significant differences regarding cognitive test performance. However, there were some significant differences regarding psychological variables as measured by the questionaries: the Ecuadorian participants had more difficulties with emotion regulation (DERS), more external locus of control (LoC), and lower self-esteem scores (RSES) compared to the migrant participants. This may reflect the different life events between the groups.

**Table 2.** Descriptive and inferential statistics comparing the national and migrant participants for demographic and psychological factors (pre-intervention).

|  | National | Migrant | Sig. | Effect Size |
|---|---|---|---|---|
| Age | 23.21 (5.00) | 24.97 (4.16) | 0.175 | 0.41 |
| Women | 8/14 (57%) | 22/35 (63%) | 0.477 | 0.053 [a] |
| Mestizo | 10/12 (83%) | 28/32 (88%) | 0.529 | 0.054 [a] |
| GHQ-28 | 9.14 (5.26) | 9.87 (5.75) | 0.836 | 0.13 |
| HADS | 10.57 (5.65) | 12.94 (5.65) | 0.196 | 0.43 |
| DERS | 84.79 (17.08) | 72.48 (16.87) | 0.025 | 0.74 |
| RSES | 30.64 (4.36) | 34.62 (4.50) | 0.005 | 0.91 |
| GSE | 21.00 (5.68) | 24.12 (4.74) | 0.058 | 0.63 |
| LoC | 9.86 (2.54) | 7.41 (2.75) | 0.006 | 0.93 |
| Trails | 2.16 (0.50) | 2.04 (0.61) | 0.283 | 0.21 |
| Hayling | 3.64 (3.84) | 3.60 (3.18) | 0.206 | 0.01 |
| Counting Span | 5.43 (1.15) | 5.57 (1.17) | 0.670 | 0.12 |

[a] = effect size given as Cramer's V, all other effect sizes are Cohen's d. GHQ-28 = General Health Questionnaire-28 total score, HADS = Hospital Anxiety and Depression Scale total score, DERS = Difficulties in Emotion Regulation Scale total score, RSES = Rosenberg Self-esteem Scale total score, GSE = General Self-Efficacy Scale total score, LoC = Locus of Control total score, Trails = Trail Making Test, ratio of switching trial to non-switching trial completion times, Hayling = Hayling Sentence Completion Test (Part B) error score.

### 3.2. Effectiveness of the Intervention

Of the 51 individuals who participated in the study, all had missing data, usually due to not completing the evaluation at the post-intervention stage. The highest data loss was for the executive function tests, in which data were missing for nine individuals (17%), and was lowest for the RSES, in which both pre- and post-intervention data were only missing

for three individuals (6%). The actual number of individuals in each pre-post comparison is indicated in Table 3, which also shows average scores for all pre-and post-intervention assessments and the results of null-hypothesis tests of the differences.

**Table 3.** Descriptive and inferential statistics for the pre-post comparison for all assessments.

| Measure | *n* | Mean Pre-Score (+SD) | Mean Post-Score (+SD) | Sig. | Effect Size *d* |
|---|---|---|---|---|---|
| GHQ-28 | 44 | 9.70 (5.58) | 5.91 (5.69) | <0.001 | 0.68 |
| HADS | 46 | 12.52 (5.75) | 8.93 (5.28) | <0.001 | 0.66 |
| DERS | 47 | 76.37 (17.75) | 69.91 (19.32) | 0.002 | 0.35 |
| RSES | 48 | 33.19 (4.73) | 34.48 (5.23) | 0.044 | 0.26 |
| GSE | 47 | 23.21 (5.23) | 25.19 (4.48) | 0.005 | 0.41 |
| LoC | 48 | 8.25 (2.90) | 7.62 (3.22) | 0.099 | 0.21 |
| Trails | 42 | 2.03 (0.50) | 2.12 (0.66) | 0.566 | 0.16 |
| Hayling | 42 | 3.41 (3.08) | 1.81 (2.04) | 0.002 | 0.62 |
| Counting Span | 42 | 5.62 (1.19) | 5.57 (1.15) | 0.736 | 0.04 |

GHQ-28 = General Health Questionnaire-28 total score, HADS = Hospital Anxiety and Depression Scale total score, DERS = Difficulties in Emotion Regulation Scale total score, RSES = Rosenberg Self-esteem Scale total score, GSE = General Self-Efficacy Scale total score, LoC = Locus of Control total score, Trails = Trail Making Test, ratio of switching trial to non-switching trial completion times, Hayling = Hayling Sentence Completion Test (Part B) error score.

From the table, we can see that as a group, the people who completed the intervention showed significantly less psychological distress afterward, measured with either the HADS or the GHQ-28, and in both cases, the effect sizes would be considered 'medium-sized'. They also reported significantly fewer difficulties with emotion regulation as measured by the DERS scale, but this is technically a 'small' effect size. Similarly, post-intervention, participants scored slightly higher for self-esteem; however, this was not much more than a one-point increase in scores on the RSES questionnaire and was qualitatively a 'small' effect. Scores on self-efficacy showed only a two-point increase from pre- to post-intervention assessment, but was nevertheless a significant increase, albeit a qualitatively 'small' effect. In contrast, there was no significant change in Locus of Control scores.

Next, we examined differences in executive function tests scores associated with the intervention. Although there was no significant change in cognitive switching ability (Trail Making Test), nor in working memory (Counting Span Test), verbal response suppression (Hayling Test) improved significantly, which had a 'medium' effect size.

When people are tested on the same assessment twice, there is the possibility that lower scores in the second assessment will be produced by practice effects. Indeed, the principle mental questionnaire measured used here, the GHQ-28, has been shown to have a large repeat-testing effect. Ormel et al. (1989) compared scores on the GHQ-28 completed on two separate dates by a group of psychiatric outpatients for whom psychological condition, as assessed by clinical interview, had not changed. Despite there being no clinical change, scores on the GHQ-28 were almost 19% lower on the second completion.

This repeat-testing effect could possibly explain our significant reduction in GHQ-28 scores. To assess the risk of this, as an exploratory measure, we increased all of the post-assessment GHQ-28 scores by the same level of practice effect reported by Ormel et al. (1989). This increased the mean post-assessment score to 7.29 (SD = 7.02), which is now closer to the pre-intervention mean of 9.70 (5.58). Nevertheless, the lower scores post-intervention compared to pre-intervention remained statistically significantly different, $F(1,43) = 9.044$, $p = 0.004$, $d = 0.38$.

We were not able to assess the other self-report measures in this way, as estimates of the magnitude of retest effects are not available, but if the reduction in psychological

distress observed can be attributed to the action of the intervention, it seems reasonable that the other changes in scores can also be interpreted in that way.

In the Hayling data presented in Table 2, error scores are analyzed, which is the normal method with this test. However, as per test instructions, response times (RT) were also recorded. The RT data mirrored that of the error score data. The participants performed faster after the intervention (mean RT = 1.22 s, SD = 0.62) compared to their performance before the intervention (mean RT = 1.92 s, SD = 0.88). This was a significant improvement, $F(1,41) = 43.169$, $p < 0.001$, $d = 0.93$.

Practice effects on RT tasks are known to occur, which could explain the improvement rather than being caused by the intervention. Nevertheless, on 15 trial RT assessments, almost all the practice effect occurs in the first five or six trials, and post hoc removal of those trials can be used to achieve equivalence between first and second test administration (Collie et al. 2003), therefore removing the practice effect. We found that the improvement in the Hayling performance remained even when we removed the first six trials from each administration. Performance was still significantly faster after the intervention (mean RT = 1.34 s, SD = 0.83) compared to before the intervention (mean RT = 2.01 s, SD = 1.04), $F(1,41) = 17.985$, $p < 0.001$, $d = 0.721$. A practice effect, therefore, seems unlikely as the cause of improved verbal response suppression after the intervention.

Finally, our sample was somewhat older than other described samples of NEET individuals, whereas many definitions define NEET as being aged 18 to 24. Only 30/49 61% of our sample were within that range, the other cases, $n = 19$, being older (range 26–36). To assess whether our results may be biased by our relatively older sample, we repeated all the analyses described above but limited the analyses to individuals under the age of 25. For most of the results, this did not change the significance of the comparisons, the differences that are shown as significant in Table 2 remained significant, and the differences that were not significant in Table 2 remained not significant. The only exception was for emotion regulation scores (DERS), which were significantly different in the full sample, but not significantly different pre- and post-intervention in the age 18–24 subsample (pre-intervention mean = 79.25, SD = 18.53 compared to post-intervention mean = 74.04, SD = 22.06), $F(1,27) = 2.143$, $p = 0.155$, $d = 0.260$).

## 4. Discussion

This study examined the effectiveness of a multicomponent intervention that focused on providing a variety of skills to prepare a diverse group of NEETs in Ecuador, which included refugees, to successfully re-engage in employment or further their education. We found that after our intervention, the group reported experiencing significantly less psychological distress, higher self-efficacy, and self-esteem and had better cognitive response inhibition than at baseline. A strength of our study compared to previous studies is that we used several outcome measures, including cognitive performance.

Improvement in mental health and associated measures may be an important consequence of the intervention. More specifically, in the context of the aims of the intervention, the reduction in psychological distress after completing the intervention may assist individuals to move into education or employment. A similar case could be made for the observed improvements in self-esteem and self-efficacy. Previous studies have focused on educational program enrolment, re-employment, and salary as the primary outcome measures (Alegre et al. 2015). For example, the main outcome measure of a systematic review that evaluated the outcomes of 18 multicomponent intervention studies saw an increase in the employment rate for those that received the intervention compared to controls (Mawn et al. 2017).

Although there is research indicating that NEETs are at a greater risk of mental health difficulties (Gutiérrez-García et al. 2017), few studies focus on health outcome measures (Nafilyan and Speckesser 2014). One study focused on providing intervention for improving health behaviors in NEETs and found that those that received the intervention were more likely to visit a psychologist (Robert et al. 2019). As with our study, other

intervention studies have reported an increase in NEET self-efficacy (Seddon et al. 2013). Other studies have also identified self-efficacy as an important mediator of re-engagement, specifically linked to more job search activity (Fort et al. 2011).

Using different outcome measures can help us further understand underlying mediators for change in this population. Indeed, one study found that excluding non-employment-focused outcomes underestimates the beneficial impact of these interventions, for example, on health (Nafilyan and Speckesser 2014). Our study adds to the existing literature by examining other outcomes measures, including emotional, psychological, and cognitive measures.

The improvement of executive cognitive functions may be of particular interest, as these are seen as core abilities that allow people to control their behavior and succeed in dynamic environments, such as in formal education (Pluck et al. 2016, 2019) or the workplace (Pluck et al. 2020b). In particular, we observed an improvement in verbal response suppression ability, as measured with the Hayling Test. This test has previously been shown to be a better predictor of academic performance in college than a standard intelligence test (Pluck et al. 2016) and predicts academic performance at the high school level (Pluck et al. 2019). Additionally, it predicts the performance of salesmen (Pluck et al. 2020b). Verbal response suppression ability is therefore emerging as a possible cognitive ability closely related to real-life achievement. The observed improvement in response suppression in our NEET sample following the intervention may therefore have practical implications.

Interventions for NEET re-engagement have varied significantly in their focus and scope. Some interventions have focused on changing educational and employment policies to reduce the number of NEETs, while others have focused on NEET prevention with strategies such as individual support (Hutchinson et al. 2016).

Many intervention studies have focused on specific skill training, and others, such as described in the current study, have used a multicomponent approach. For example, previous interventions on NEETs have focused on delivering support, internship placements, vocational, social, and education. The intervention in this study has many of these elements, specifically: industrial quality and productivity, industrial techniques, sales and customer service, sports activities, audiovisual production, financial education, customer services for hotels and restaurants, entrepreneurship, communication, psychology, and job search skills. A systemic review found that more intensive programs with multiple components had a greater impact on employment rate and salary in the long term (Mawn et al. 2017).

A multicomponent intervention approach might be better able to provide skills and support for this group, as it can provide a variety of skills for a group with varying needs. Indeed, the NEETs that participated in our study are a heterogeneous group, including refugees from different countries. This is consistent with what has been found in previous studies. 'NEET' in general has always been a global label of young people without employment and not enrolled at school (Furlong 2006; Serracant 2014).

Furthermore, there is research to suggest that it is also important to adapt the intervention to fit the population. Indeed, a recent systematic review indicated that tailoring interventions to specific cultures and minority groups improved outcomes; this includes therapist training, the content of therapy, and how the services are delivered (digitally, face-to-face, over the telephone, group, individual, length of intervention; Arundell et al. 2021). The results of this review paper further highlight the importance of multicomponent interventions that can adapt to meet a specific population's needs.

We also found that the national participants in the intervention (i.e., Ecuadorians) differed from the migrant participants in several psychological factors. In particular, compared to the migrant participants, the national participants reported more difficulties with emotion regulation and external loci of control, as well as lower self-esteem scores. This likely reflects the different past histories of the two groups. It would have been of interest to compare the pre- and post-intervention scores separately for these groups. However, our sample size was too small to allow it, which should be noted as a limitation of the current study.

The findings in this study need to be considered in light of several other limitations. We did not have a control group, and therefore it is possible that the apparent improvements in mental health and cognitive ability were due to test practice. Our detailed analyses suggested that this was not the case. Nevertheless, such effects cannot be fully ruled out. Another limitation is that we did not use employment or school enrolment as an outcome measure, so we do not know the extent to which our intervention could have an actual effect on employability. Similarly, we cannot determine to what extent this intervention had an enduring effect on participants. Our intervention had multiple components, and our sample was very heterogeneous, and this prevented us from ascertaining which components are essential for change and which are less necessary for subgroups within our NEET sample.

Despite these limitations, this study contributes to our understanding of NEETs in Latin America and potential interventions that can assist this vulnerable population to re-engage in education and employment. However, further research is needed to establish which components of this intervention are most effective for particular subgroups of NEETs and if this intervention could be effective for other cultures or different contexts than that of this study.

**Author Contributions:** Conceptualization, C.C.-A. and A.F.T.; methodology, A.F.T. and G.P.; formal analysis, G.P.; investigation, C.C.-A.; data curation, C.C.-A. and G.P.; writing—original draft preparation, C.C.-A., A.F.T. and M.S.G.; writing—review and editing, M.S.G.; project administration, C.C.-A. All authors have read and agreed to the published version of the manuscript.

**Funding:** This research received funding from the Articles Publication Fund of Universidad San Francisco de Quito USFQ.

**Institutional Review Board Statement:** The study was conducted following all local laws and guidance on research provided by the American Psychological Association and in accordance with the Declaration of Helsinki of 1975, revised in 2013 and approved by the Bioethics Commit-tee of Universidad San Francisco de Quito reviewed the protocol, including the pre- and post-intervention assessments (application code: 2018-098IN, approval granted 9 May 2018).

**Informed Consent Statement:** Informed consent was obtained from all subjects involved in the study.

**Data Availability Statement:** The data that support the findings of this study are available from the authors. The data has not been placed in a public repository because of restrictions with the ethics committee approval.

**Acknowledgments:** We would like to thank Ana María Viteri, Gabriela Romo, Raquel Galindo, María Gloria Medina, and Isabela Lara who assisted with management and data collection on this research.

**Conflicts of Interest:** The authors declare no conflict of interest.

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
