# Peer review of "Multicomponent Intervention Associated with Improved Emotional and Cognitive Outcomes of Marginalized Unemployed Youth of Latin America"

_socsci, doi:10.3390/socsci11040155_

Round 1
Reviewer 1 Report
Thank you for inviting me to review this interesting and informative study investigating the outcomes of a multicomponent intervention in a group of adolescent and adult refugees aimed at supporting them into education, training, or work. The study has many merits and could make an important contribution to the literature in this area, and I imagine it will be of considerable interest to a large number of readers. There are however a number of important limitations to the study and the manuscript itself, outlined below. It is my opinion that these need to be addressed before the manuscript will be suitable for publication.
Major Comments
- I don’t understand how the inclusion criteria for the intervention match up to the suggestion that this was a study about an intervention for those that are NEET. In the introduction you define NEET status as meaning someone is aged 15-24 years old and not in education employment or training. A number of other studies adopt the WHO definition for young adults i.e. those aged 18-24 (Walker-Harding, Christie, Joffe, Lau, & Neinstein, 2017), but regardless, this study included adults up to the age of 36. Firstly, can the authors describe how many of their sample were older than the upper limit typically used to define NEETs. Secondly, can they either justify the inclusion or a larger age range for NEETs in this study, or can they ensure the description of who this intervention was for (in the introduction and methods) makes it clear that it was for adults up to the age of 36 and not just younger adults. Thirdly, given the mean age was nearly 25 it might perhaps be best to present a subgroup analysis for those aged <25 that would ordinarily be considered NEETs even if a different age category was used to define NEETs in the present study.
- The focus on cognitive measures need to be more thoroughly justified. The point that improvement of these cognitive functions is either possible or important to the health and lives of the target population is not explained well enough, e.g. are these indicators of employability, educational attainment, or acceptance on to educational or training courses? Without such a justification, it is hard to understand the relevance of the focus on cognitive outcomes and how this is a strength of the study.
- The authors make a number of references to hypotheses in the methods section, e.g. lines 216-218 “We, therefore, used the total DERS score to test our principal hypothesis regarding emotional regulation skills.” The hypotheses are not stated in the manuscript. This may just be a semantic issue, perhaps the word hypothesis in the methods refers to outcomes instead as it seems that the scores on each measure was used as a way of evaluating the intervention. If that is the case, please change the wording so that these dependent variables/outcomes are not referred to as hypotheses. However, if the authors did mean to say that they had a number of hypotheses (i.e. predictions about what effects might be found in this study) then can they please explicitly state the hypotheses prior to any statement of results in relation to them and note if the hypotheses were developed prior to conducting the study, in the process of doing so, or post hoc.
- Can the authors more thoroughly justify their choice of non-parametric tests for non-normally distributed variables. Firstly, not all parametric models require normally distributed data, it is often more important to consider the distribution of the residuals. Secondly, it is often the case that transformations to the data can lead to approximations to the normal distribution which are appropriate to allow parametric tests to be used. Parametric tests are favored as they have far greater power to detect effects and given the small sample size here this would seem important.
- I fail to see the relevance of presenting data on the internal consistency and the validity of the scales. The authors have used validated measures with the psychometric properties studied in much larger samples than used in the present study, and the aims of this study did not necessitate the presentation of data on reliability and validity as tested in the present (small) sample. Therefore, in my opinion, these sections of the manuscript can be removed to save words and focus instead on those results that are of greater relevance to the aims of the study.
- The authors do not need to make adjustments for multiple testing, particularly if they have followed a hypothesis-driven analysis – see (Perneger, 1998; Rothman, 1990). It would be more informative for the authors to focus on the context and detail of the results rather than on the arbitrary determinations of “statistical significance”. It is of course important to consider the potential for their results to be due to random error, but presenting exact p-values and discussing this as a small component part of the evaluation of the intervention would be more informative to the reader.
- The Cohen’s d for DERS and RSES appears to be incorrectly calculated. In Table 2 the authors present a Cohen’s d of 7.50 for DERS and 2.00 for RSES, these are the differences between the respective pre-post means (i.e. pre-mean – post-mean), without any division by the pooled standard deviation.
- The tests for regression to the mean, and generally the comparison to another small study to infer something about the current sample and the effects found does not seem appropriate. Both the current sample and the reference sample are incredibly small, it is therefore difficult to imagine that either group are generally representative of the apparent target population (young adults in Quito) and so there seems little to learn from these analyses. Instead, it might be better to find a study that presents some information on the degree of change expected by chance on the GHQ-28 or HADS, or some consideration of minimal clinically important differences on these measures (Bauer-Staeb et al., 2021; Button et al., 2015; Kounali et al., 2020). This would then allow the authors to consider if the degrees of change found in this study are potentially 1) of a magnitude unlikely to be due to chance alone, and 2) of a magnitude that might be clinically important or meaningful for patients. This would not answer the question of whether or not regression to the mean occurred here (it is very likely that it did to some degree), instead it gets to the more important point which is: is it likely that the whole of the effect found here is due to regression to the mean, and if it isn’t might it be the case that the effects represent a potentially clinically important difference that patients might find meaningful.
Minor Comments
- The authors make some very interesting and important points about the mental health of those that are NEET but it may be important to also inform the reader that recent studies have demonstrated that NEETs not only have poorer mental health, they also appear to have worse outcomes from mental health treatments (Buckman et al., 2021). This is the next step of relevance given the intervention in this study.
- In the discussion of refugees, given the interventional focus of this study it might be helpful to touch on the literature regarding adapting treatments to enhance the probability of positive outcomes for refugees. A recent systematic review and meta-analysis demonstrated that positive outcomes come not only from culture-specific adaptations, or adaptations to the content of treatment but also the organisation of services and structures seeking to support the target population (Arundell, Barnett, Buckman, Saunders, & Pilling, 2021). You might argue that the multicomponent intervention used in the present study taps into several of the domain discussed in the typology presented in that review paper.
- The statement of the “aims of the study” on lines 136-139 summarize one of the main findings and implications of the study. Such statements would seem more suited to the Discussion section as they do not relate to the aims of the study.
- The descriptive statistics of the sample should be moved from the methods section to the start of the results section, and the authors should present absolute figures not just percentages.
- Please provide a list of abbreviations and their meanings as a footnote to Table 1.
References
Arundell, L.-L., Barnett, P., Buckman, J. E. J., Saunders, R., & Pilling, S. (2021). The effectiveness of adapted psychological interventions for people from ethnic minority groups: A systematic review and conceptual typology. Clinical Psychology Review, 88(June), 102063. https://doi.org/10.1016/j.cpr.2021.102063
Bauer-Staeb, C., Kounali, D., Welton, N. J., Griffith, E., Wiles, N. J., Lewis, G., … Button, K. S. (2021). Effective Dose 50 method as the Minimal Clinically Important Difference: Evidence from depression trials. Journal of Clinical Epidemiology. https://doi.org/10.1016/j.jclinepi.2021.04.002
Buckman, J. E. J., Stott, J., Main, N., Antonie, D. M., Singh, S., Naqvi, S. A., … Saunders, R. (2021). Understanding the Psychological Therapy Treatment Outcomes for Young Adults Not in Education, Employment, or Training (NEETs), moderators of outcomes, and what might be done to improve them. Psychological Medicine. https://doi.org/10.17605/OSF.IO/7RQWD
Button, K. S., Kounali, D., Thomas, L., Wiles, N. J., Peters, T. J., Welton, N. J., … Lewis, G. (2015). Minimal clinically important difference on the Beck Depression Inventory - II according to the patient’s perspective. Psychological Medicine, 45(15), 3269–3279. https://doi.org/10.1017/S0033291715001270
Kounali, D., Button, K. S., Lewis, G., Gilbody, S., Kessler, D., Araya, R., … Lewis, G. (2020). How much change is enough? Evidence from a longitudinal study on depression in UK primary care. Psychological Medicine, 1–8. https://doi.org/10.1017/S0033291720003700
Perneger, T. V. (1998). What’s wrong with Bonferroni adjustments. BMJ, 316(7139), 1236–1238. https://doi.org/10.1136/bmj.316.7139.1236
Rothman, K. J. (1990). No adjustments are needed for multiple comparisons. Epidemiology, 1(1), 43–46. https://doi.org/10.1097/00001648-199001000-00010
Walker-Harding, L. R., Christie, D., Joffe, A., Lau, J. S., & Neinstein, L. (2017). Young Adult Health and Well-Being: A Position Statement of the Society for Adolescent Health and Medicine. Journal of Adolescent Health, 60(6), 758–759. https://doi.org/10.1016/j.jadohealth.2017.03.021
Reviewer 2 Report
There are two components to this — reasons that the youth and young adults are without work, etc. and the way they intervention can help. Both are interesting and relevant given the high rates of migration in Latin America.
The topic sits within a larger body of research on the adjustment of migrants, and offers an interesting perspective on this by examining how an intervention might help their mental health. What makes it all the more significant is the emphasis on nations and populations that and who are not always the focus of research on migration.
The conclusions follow based on the analysis done. I recommend going a step further to examine questions of how immigrants from different backgrounds and ages responded.
This is a very well written discussion of the intervention and statistical results. One question that emerges after the context focuses on identifying migration as a factor, and breaks down the research population by country of origin or age, and how and if that factors into the responses and your results. It is suggested that you integrate the issues of identity into your analysis and discussion.
Overall, it should be published after minor revision.
